# Lactoferrin as a Human Genome “Guardian”—An Overall Point of View

**DOI:** 10.3390/ijms23095248

**Published:** 2022-05-08

**Authors:** Iwona Bukowska-Ośko, Dorota Sulejczak, Katarzyna Kaczyńska, Patrycja Kleczkowska, Karol Kramkowski, Marta Popiel, Ewa Wietrak, Paweł Kowalczyk

**Affiliations:** 1Department of Immunopathology of Infectious and Parasitic Diseases, Medical University of Warsaw, 02-106 Warsaw, Poland; ibukowska@wum.edu.pl; 2Department of Experimental Pharmacology, Mossakowski Medical Research Institute, Polish Academy of Sciences, 02-106 Warsaw, Poland; 3Department of Respiration Physiology, Mossakowski Medical Research Institute, Polish Academy of Sciences, 02-106 Warsaw, Poland; kkaczynska@imdik.pan.pl; 4Military Institute of Hygiene and Epidemiology, 4 Kozielska St., 01-163 Warsaw, Poland; hazufiel@wp.pl; 5Maria Sklodowska-Curie, Medical Academy in Warsaw, Solidarnosci 12 Str., 03-411 Warsaw, Poland; 6Department of Physical Chemistry, Medical University of Bialystok, Kilińskiego 1 Str., 15-089 Bialystok, Poland; kkramk@wp.pl; 7Department of Animal Nutrition, The Kielanowski Institute of Animal Physiology and Nutrition, Polish Academy of Sciences, Instytucka 3, 05-110 Jabłonna, Poland; marta.popiel@nutropharma.pl; 8R&D Department Nutropharma Ltd., Jedności 10A, 05-506 Lesznowola, Poland; ewietrak@nutropharma.pl

**Keywords:** lactoferrin, oxidative stress, DNA damage, DNA glycosylases

## Abstract

Structural abnormalities causing DNA modifications of the ethene and propanoadducts can lead to mutations and permanent damage to human genetic material. Such changes may cause premature aging and cell degeneration and death as well as severe impairment of tissue and organ function. This may lead to the development of various diseases, including cancer. In response to a damage, cells have developed defense mechanisms aimed at preventing disease and repairing damaged genetic material or diverting it into apoptosis. All of the mechanisms described above are part of the repertoire of action of Lactoferrin—an endogenous protein that contains iron in its structure, which gives it numerous antibacterial, antiviral, antifungal and anticancer properties. The aim of the article is to synthetically present the new and innovative role of lactoferrin in the protection of human genetic material against internal and external damage, described by the modulation mechanisms of the cell cycle at all its levels and the mechanisms of its repair.

## 1. Introduction

Lactoferrin (LF) is a glycoprotein belonging to the group of transferrins, i.e., proteins capable of binding and transferring iron ions. It was first isolated from cow’s milk in 1939. In 1960, 3 independent laboratories confirmed that LF is the main iron-binding protein in human milk. Intriguingly, human and beef LF are characterized by very high compatibility in terms of structure (77%), while the functionality is practically identical, therefore, due to easier extraction, beef LF is used for research and added to food products and dietary supplements [1,2,3,4]. LF is present in milk of various species of mammals, including cows, pigs, mice, and humans [5,6,7,8]. Its major functions are antioxidant, antimicrobial and anti-inflammatory [9,10]. Moreover, it has been postulated as a biomarker in the diagnosis of certain diseases such as inflammatory bowel disease (IBD), Alzheimer’s disease (AD) and dry eye disease (DED) [11,12,13]. LF multifunctionality is a result of interactions with variable hosts’ and microbial molecules (such as: cell receptors, glycosaminoglycans—GAGs, lysozyme, nucleic acids, lipoteichoic acid—LTA, lipopolysaccharide—LPS) as well as iron deprivation and nuclear DNA binding [14,15,16,17,18,19,20,21,22,23,24,25]. Many studies showed that LF acts as a transcription factor [26] triggering the expression of a variety of genes, including genes related to the innate immune response [27], lipid metabolism (fatty acid oxidation, elongation, synthesis and degradation) [28,29], and heterogeneous metabolism and lysosomal degradation [30,31]. Nonetheless, the protective role of LF against human DNA damage resulting from environmental, biological and physiological conditions is still poorly documented. The proper balance between DNA repair, proliferation, and apoptosis prevents mutations accumulation within human genome, genomic instability and in consequence development of a broad spectrum of diseases. This article is an attempt to overview the multitude of DNA damaging factors and answer the question of how lactoferrin can influence these processes and protect host genetic material, in overall perspective.

## 2. DNA Damage

DNA carries the information necessary for the structure and function of cells, thus DNA damages are a serious problem for the organism. DNA damage can lead to mutations and the latter can cause various diseases, including cancers. Damage to DNA is a change in its structure, such as modification of a nucleotide, or interruption of both or one strand of DNA. Replication of DNA with a damaged nucleotide results in the insertion of an incorrect nucleotide in the complementary strand. Breaking both DNA strands can also result in incorrect repair and lead to a change in the DNA sequence. These changes can lead to damage to the structure of the chromosome, alter the information in the gene, cause the gene to malfunction or lead to its dysregulation. DNA damage in proliferating cells often leads to carcinogenesis, whereas DNA damage in non-dividing differentiated cells results in the accumulation of abnormalities, blocked transcription, reduced gene expression and premature ageing and cell death [32]. Many exogenous and endogenous factors can cause DNA damage [33,34,35,36]. 

### 2.1. DNA–Damaging Factors

The DNA of mammalian cells, including humans, is constantly exposed to various internal (lipid peroxidation induced by oxidative stress) and external factors (e.g., Roentgen radiation, UV radiation, chemical compounds present in the air and inhaled) that may affect its modifications. DNA damage, that ultimately can lead to mutations and genomic instability, is due to many factors, such as oxidative stress, metabolic disorders, viral and microbial pathogens, excess cellular proliferation and chemical factors. 

Numerous literature data indicate that also mucochloric acid present in drinking water, soft drinks, oxygen and nitrogen from the air, and methylation factors can contribute to the modification of nucleic acids in all living cells [35,36]. Unrepaired DNA lesions (adducts) can lead to various types of pro-inflammatory morphological changes in the cell structure, including permanent and irreversible damage to mitochondria leading to cell death, various types of carcinogenesis, aging, and a wide spectrum of neurodegenerative diseases such as Wilson’s, Alzheimer’s and Parkinson’s diseases [37]. Exocyclic DNA adducts, which include 1,N^6^-ethenoadenine (ɛA), 3,N^4^-ethenocytosine-(ɛC) and N^2^,3-ethenoguanine-(ɛG), show mutagenic, teratogenic and clastogenic effects in *E. coli* as well as mammalian cells [34,35,37,38,39] (Figure 1).

#### 2.1.1. Chemical Factors

The external factors modifying DNA include: gamma and beta radiation, X-rays, various carcinogenic chemical compounds—vinyl chloride (VC) or chloroacetaldehyde (CAA) or drugs taken orally. These compounds, especially active ingredients of drugs, participate in the cellular metabolism leading to DNA damage in the cell. Moreover, they can also induce inflammation of precancerous lesions in tissues [34,35,36,40]. 

Also tobacco smoke is known to contain over 5000 substances, of which over 150 show toxicity, together with mutagenic and/or carcinogenic effects. The most potent carcinogens in tobacco smoke include acrolein, acrylonitrile, acetaldehyde, formaldehyde and ethylene oxide [41]. Repair processes play a key role in preventing carcinogenesis and the development of other diseases resulting from DNA damage. Therefore, of great concern is the occurrence of mutations in genes encoding proteins that recognize damaged genetic material, activate the repair process, or repair damaged DNA. Specifically, the genotoxic properties of chemical agents disrupt the mechanisms of base excision repair (BER) and nucleotide excision repair (NER) [35,36,39]. The most common causes of inactivation of these important genes include point mutations within the gene, deletions, insertions, gene conversion and epigenetic silencing [42]. This leads to impaired DNA repair and accumulation of abnormalities resulting in the above-mentioned outcomes.

#### 2.1.2. Biological Factors

DNA damage may also occur during viral and intracellular microbial infections as a result of pathogen mediated genotoxicity or pathogen interference with host cell repair processes [43]; (Figure 2). The host cellular pathways responsible for DNA repair could be impaired by microbial proteins resulting in accumulation of mutations, genome instability and finally increase risk of cancer development. Moreover, microorganisms may modulate gene expression, activate oncogens or inactivate tumor suppressor genes favouring cellular transformation [44]. The best well known DNA and RNA viruses associated with cancers are HTLV-1, HPV, HBV, HCV, EBV, and KSHV [43]. They modify cell cycle function, dysregulate DNA damage signalling, and repair pathways (DDR kinases) inducing tumor development, supporting tumor promotion and spreading [45,46,47,48]. The mechanism leading to DNA damage as a result of human papillomavirus infection is very interesting. HPV reduces *p53* expression levels during reactive nitrogen species (RNS) stress, leading to an increase in DNA damage [48].

Moreover, host cell’s DNA may undergo damage upon several bacterial infections such as *Helicobacter pylori* [49], *Listeria monocytogenes* [50], *E.coli* [51], *Mycoplasma fermentans* [52], *Shigella* [53] and *Neisseria gonorrhoeae* [54]. Handa and colleagues showed that *Helicobacter pylori* infection induces increased production of oxygen free radicals and RNS in the stomach damaging the DNA of gastric epithelial cells [49]. The mechanisms of bacterial action include enhancement of DNA-damage accumulation, and also inhibition of *p53*-activities and/or cell cycle arrest [55] influenced through toxins, or due to activation of the immune response [56,57]. Lipopolysaccharides, major components of the outer membrane of Gram-negative bacteria, increase intracellular ROS levels through mitochondrial derived NADPH oxidase, inducing macrophages and neutrophils. Mitochondrial dysfunction is the result of mtDNA depletion and inhibition of mitochondrial transcription. Additionally, LPS increases the level of 8-oxo-7,8-dihydro-2′-deoxyguanosine (8-oxoG), biomarker of DNA oxidative damage, in nuclear DNA in the same way as H_2_O_2_ [58].

#### 2.1.3. Physiological Conditions

Numerous natural processes in the organism including breathing and digestion leads to production of harmful free radicals contributing to DNA damage. In addition, an unhealthy lifestyle including smoking, alcohol consumption, poor diet, or exposure to other harmful environmental factors can exacerbate oxidative stress [59]. According to “oxidative stress theory” of aging the increase in ROS is closely associated with the aging process and leads to functional changes and pathological states e.g., neoplastic transformation or neurodegenerative diseases [60,61,62]. Both nuclear and mitochondrial DNA damage induced by oxidative stress that are additionally linked with age-related metabolic changes in the human body, among others, accumulates over time. DNA damage and mutations, a consequence of faulty DNA repair, have been confirmed to accumulate with age and their consequences progressively impede cellular function and increase vulnerability to the development of chronic disorders associated with aging [61,63,64].

### 2.2. Oxidative Stress and Repair Mechanisms

Oxidation of polyunsaturated fatty acids through ROS in the so-called lipid peroxidation results in the formation of several main products, which include: malondialdehyde (MDA) and trans-4-hydroxy-2-nonenal (HNE), crotonaldehyde and 2,3-epoxy-4-hydroxynonanal [65]. The resulting derivatives can strongly modify DNA and inactivate functional DNA repair proteins from the group of glycosylases by sequential cross-linking, thus affecting important metabolic pathways in the body [65]. Under typical conditions of cell growth and development, the excessive production and neutralization of ROS also depends on the action of antioxidant enzymes, which include: superoxide dismutase, catalase and glutathione peroxidase, xanthine oxidase and repair enzymes of the BER and NER pathway, TLS and the SOS system. Deficiency of these enzymes in the cell disrupts its homeostasis and leads to the excessive production of oxygen radicals, including the superoxic and hydroxyl radical anion in the Haber-Weiss-Fenton reaction [66,67,68]. A typical example is cells exposed to UV radiation, showing an excessive level of ROS-induced damage through DNA modification [69,70] and the secretion of pro-inflammatory cytokines [71,72]. An example of a modification in genomic DNA is 2′-deoxyguanosine, which leads to damage in the strand to oxidized 8-hydroxy-2′-deoxyguanosine (8-OHdG) [70]. 8-OHdG, considered a new and typical biomarker of DNA oxidative damage similar to the CRP protein, is a modification recognized by several repair systems, such as BER, NER or Mismatch Repair, TCR, with the participation of the Fpg protein encoded by the OGG1 enzyme [69,71]. 

## 3. Lactoferrin

Human LF is highly multifunctional protein that reflects several enzymatic activities. It is able to bind and hydrolyse DNA [20,21,22,23,24] as well as RNA [21,25,32,33], acts as a specific transcription activator of DNA sequences [6,23], binds to various nucleotides showing ATPase and phosphatase activities [6,21,34,35]. Lactoferrin has also been shown to bind oligo- and polysaccharides with amylase activity [6,73]. Moreover, it possesses cytotoxic properties and induces apoptosis [21]. LF plays an important function in the homeostasis of the entire immune system, reduces oxidative stress at the molecular level as well as local inflammation [6,7,8]. The multiplicity and variety of LF properties makes it could be referred to as the “guardian of the genome” (Figure 2).

The beneficial effects and a potential clinical utility of LF different types: bovine LF (bLF), human LF (hLF), recombinant hLF (rhLF), intracellular delta-lactoferrin (delta LF), conformational states: apo-LF (open iron-free form) and holo-LF (closed iron-binding form), and fragments: lactoferricin (the N-terminal region of lactoferrin) are investigated [74]. The majority of research concern on bLF properties that shares a 69% amino acid sequence homology with hLF [75] but reflects differences in glycosylation pattern (determining protein structure, biological function, stability and influence immunogenicity and antigenicity [76,77,78,79]. Wherefore for clinical use rhLf has been expressed in different expression systems and cell types: BHK cells, Sf9 insect cells CHO cells; plants: rice; fungi: *Aspergillus niger*; yeast: *Pichia pastoris*; and transgenic animals (cow, goat, sheep, mouse) [80,81,82,83,84]. Interestingly, delta LF acts as a transcription factor. It adhere to the cell membrane and, after internalisation from the cell surface, localize within nucleus [85] as well in the cell’s cytoplasm [86,87,88,89]. Liu and colleagues suggest that delta LF protein is formed by expression of the *LF* gene from the alternative P2 promoter, resulting in a 73 kDa intracellular delta LF protein [88]. 

### 3.1. Lactoferrin as a Human Genome “Guardian” (Indirect and Direct Action)

LF role as DNA protective factor may be dual and results from its indirect or direct action on human genome (Table 1). 

Elimination of DNA damaging factors (e.g., hydroxyl radicals, microbes, up-regulated immune cells) or consequences (e.g., cancer cells), without interaction with host genetic material, could be pointed as an indirect mechanism of DNA protection by LF. These mechanisms included iron saturation, scavenging hydroxyl radicals, cell cycle modulation, antimicrobial and anti-inflammatory function.

On the other hand its direct action concern on the ability to DNA binding, hydrolyzation of nucleic acids as well as degradation instead of DNA [6,73,91]. All direct actions of lactoferrin are included in Figure 3.

### 3.2. Examples of LF (Indirect and Direct) Functions Providing Protection of the Human Genome—An Overview

#### 3.2.1. LF as a Transcription Factor—A Similar to p53 Mode of Action

Human LF possesses two DNA binding sites that interact with nonspecific and specific DNA [140] and the ability to activate specific DNA sequences as a unique transcription factor [10]. LF action is very similar to *p53* protein through its capacity to bind the DNA and potency to stimulate the expression of a number of genes such as Bax, SelH, DcpS, UBE2E1, Skp1 and GTF2F2 and regulate cell cycle, apoptosis and cell differentiation. LF in similar way to *p53* protein can enters the cell nucleus [144,145,146] where it binds to specific DNA sequences, thus regulating the transcription of anti-inflammatory cytokine genes, genes promoting apoptotic cell death, and these genes that codes proteins showing anti-proliferating and anti-cancer activity [141,147,148,149]. In addition, LF has been shown to transactivate the *p53* through the activation of nuclear factor NF-κB and in this specific way regulate the p53 tumor suppressor gene [142]. In fact, LF acts similarly to p53, which is a good known “guardian of the genome” that helps maintain the genetic integrity of the cell after DNA damage. Lactoferrin is believed to act as a sensor for damaged DNA; once the DNA damage is extensive and unrepairable, this protein can kill the cell by inducing apoptosis, when the DNA damage is fairly minor and repairable, similarly, p53 stops the cell in the G1 phase to permit DNA repair mechanisms to work [104,107]. 

#### 3.2.2. LF in ROS Ratio Control

The protein that regulates the concentration of iron ions in plasma and transports them to tissues is transferrin (a glycoprotein). The body contains about 3 to 4 g of iron, a component necessary for key biological processes, which may be involved in the cell’s response to oxidative stress, participates in the transport of oxygen to the body’s cells, affects cholesterol metabolism and promotes detoxification and indirectly contributes to the synthesis of serotonin, prostaglandins, nitric oxide, and the production of thyroid hormones. It also participates in the synthesis of DNA and plays an important role in fighting viruses and bacteria by the immune system [39,66].

An example of transferrin is LF: which is produced mainly by the epithelium of the mucosa of mammalian cells. Therefore, it is abundant in the milk of mammals and other secreted fluids, such as tears, saliva, pancreatic juice, and is present in the granules of multinucleated leukocytes [3,22,23,24]. LF exhibits variable properties such as antioxidant, anti-inflammatory and antimicrobial. Therefore, it plays an important role in the body’s defense against infections and may exert an anti-inflammatory effect by inhibiting the formation of hydroxyl free radicals [36,37,40]. Through its antioxidant properties, LF can prevent DNA damage, thereby preventing tumor formation. A potential mechanism by which LF reduces oxidative damage at the cellular and tissue levels after LPS is modulation of iron level in the cell (Figure 4). LF has been demonstrated to possess a significant antioxidant activity due to direct iron chelation or enhancing the activity of key antioxidant enzymes such as superoxide dismutase, glutathione peroxidase (GPx), catalase, which are directly involved in ROS scavenging [90]. LF protects against oxidative damage in the mitochondria and for example indicate its potential in the prevention and treatment of systemic inflammatory response syndrome and its transition to septic conditions in vivo and the development of neurodegenerative diseases [21,33]. 

Moreover, the protective effect of LF against oxidative DNA damage has been shown to be related to scavenging ROS independent of its iron chelating activity and also to LF degradation by free radicals instead of DNA damage [91]. Interestingly, LF is more susceptible to degradation than other cow’s milk proteins. 

#### 3.2.3. Human Immune System Regulation

Several effects of LF on immune function and inflammation regulation have been described, although contrasting results were frequently reported: (1) limitation of the immune system response and reduction the risk of cell damage, e.g., in the case of infections and non-infectious disorders such as arthritis, allergies or cancer [150] and (2) stimulation of the immune response: anti-cancer and anti-microbial effect [21,96].

LF is able to support the proliferation, differentiation and activation of immune cells by direct interactions with cell surface components and to strengthen their responses. Furthermore, enhancement of the immune response may result not only from the ability of LF to remove iron deposits, but also from the modulation of cytokine expression and effects on immune cell migration and lytic and phagocytic activity [96].

##### Decrease of Inflammation and Cell Damages—Immunosuppression

The anti-inflammatory activity of bLF supports the injury improvement and prevents tissue damages during the development of inflammation [93,95,151]. Due to its antibacterial action and the binding of several pro-inflammatory pathogen associated microbial patterns (PAMPs): LPS (bacterial endotoxins) [101], unmethylated CpG-containing DNA [15]; or their receptors: the toll-like receptors (TLRs) [152], it prevents inflammation and subsequent tissue damage caused by the release of pro-inflammatory cytokines and reactive oxygen species. The interaction of LF with negatively charged groups present on the surface of the immune cells activate signaling pathways that induce a physiological anti-inflammatory reactions [96]. Moreover, by regulating gene expression e.g., by blocking NF-kB (nuclear factor kappa-light-chain-enhancer of activated B cells), lactoferrin may also modulate cytokine synthesis. It decreases the production of some pro-inflammatory cytokines, such as tumor necrosis factor α (TNF-α), interleukin 1α (IL-1α) and interleukin 6 (IL-6) or interleukin 8 (IL-8); [150,153] and increases the levels of anti-inflammatory IL-10 [94]. LF has been demonstrated to neutralize free LPS by disruption the complex of LPS-binding protein—endotoxin that activate the TLR4 signaling pathway triggering monocytes and macrophages activation [100,154,155]. During viral infections LF reduces the synthesis of chemoattractants, such as IL-8 and monocyte chemoattractant protein-1, minimalizing development of pathology [92]. Finally, by controlling oxidative stress, LF alters the production of immunoregulatory mediators, that are crucial for the induction of an adaptive immune response [156] and protects against iron disorders by modulating immune response and down-regulating pro-inflammatory cytokines, such as IL-6, in vitro and in vivo models and in clinical trials [156,157].

##### Immunostimulation

Several mechanisms are involved in the immunomodulating activity of lactoferrin [3,95,96,156,157]. It plays an important role in recruitment, activation and antigen presentation by antigen-presenting cells (APCs), such as monocytes and dendritic cells [156,158]. LF regulates maturation of DCs, increases their capacity to trigger proliferation and release IFN-α and decreases antigen internalization [158]. It activates macrophages to release pro-inflammatory molecules, e.g., TNF-α, IL-8, and nitric oxide [159], and increases their phagocytic activity [95,156]. The lactoferrin-dependent control of proinflammatory cytokine expression, represents an important mechanism that results in effects on the growth, differentiation, activation, and functions of immune cells [1,3,14,95,96,156,157]. bLF can promote activation of intestinal transcription of some essential genes such as p40, IL-12 and NOD2, essential for systemic immunity activation [150]. Furthermore, many in vitro studies reported lactoferrin as a modulator of both cell-mediated and humoral adaptive immunity triggering: (1) the synthesis of non-specific IgA and IgG antibodies (in the intestine), [160]; (2) the maturation of B lymphocytes into efficient antigen presenting cells [4], (3) activation of B, T and NK lymphocytes (in the intestine, spleen and peripheral blood) [4,115,161,162] (4) differentiation the precursor T cells present in the thymus to acquire the CD4 + CD8- Th helper cell phenotype [98,161]. LF may also enhance the cytotoxic functions of NK and lymphokine-activated killer cells, potentially through binding to RNA and DNA [115,161].

#### 3.2.4. Antitumor Action of LF

LF is able to inhibit or activate cell division and movement, depending on whether it affects cancer cells or healthy normal body cells [158,163]. Administration of LF has been shown to inhibit or even prevent tumor growth by activating the adaptive immune response or decreasing the expression of growth factor protein e.g., of vascular endothelial growth factor protein in human lung cancer cell line—A549 [164]. Moreover, it increases the expression of surface receptors on neoplastic cells, facilitating their identification by the immune system [21,97]. Li and colleagues have shown that the anti-cancer activity of lactoferrin may also be related to its ability to inhibit angiogenesis and the growth of blood vessels to the tumor [102]. Thanks to this, it reduces the size of tumors and the possibility of metastasis formation [165]. LF also has a lytic effect on cancer cells. Importantly, it acts selectively, showing greater cytotoxicity towards neoplastic cells than normal ones. Furthermore, it was shown that bovine LF exerts a cytotoxic effect on fibrosarcoma, melanoma, head and neck, as well as colon cancer cells, and inhibits the proliferation of lung cancer cells [21,102,163,164,166]. 

It has been shown that lactoferrin cytotoxicity towards cancer cells can occur through various pathways, not only by the above mentioned inhibition of angiogenesis or induction of immunoreactivity, but also through cell cycle arrest, cell membrane damage or induction of apoptosis of dangerous cells.

#### 3.2.5. Cell Cycle Arrest

In humans cell cycle is strictly regulated by hormones, cyclins, cyclin-dependent kinases and inhibitors of cyclins. It has been revealed that lactoferrin may trigger cell cycle arrest [21,102,167]. It binds to nucleic acids in the cell and through cell cycle regulation affects the balance of cell proliferative activity as well as may mediate apoptosis induction [106]. This process is cell type dependent and provides antitumor activity of LF (Figure 5), as well as likely inhibits bacterial genotoxicity.

bLF has been shown to arrest breast cancer MDA-MB-231 cells at the G2 phase, whereas MCF-7 cells at the G1 or G2 phase [149] and squamous cell carcinoma, nasopharyngeal carcinoma cells, human cancer cell line O12, and canine mammary gland adenocarcinoma cell line at the G1 phase [21,102]. Furthermore, some other studies have demonstrated that bLF retain growth of breast cancer cells at the G1 and transition of cell cycle to S phase. Lactoferrin induced cell cycle arrest at the G_1_ to S phase in due to regulation of cell cycle-associated proteins including Akt, p21, p19, p27, Cdk2, cyclin E, Cdk4, and cyclin D1 [103,104,168]. LF could modulate the expression and activity of p21^cip1^ and p27^kip1^ due to modulation of the phosphatidylinositol 3′-kinase/Akt pathway [103,169]. The decrease of phospho-Akt may be a result of altered availability or stability of insulin-like growth factor after LF binding [170]. 

It has been revealed that bLF, in the MDA-MB-231 cells, was able to increase the level of cyclin-dependent kinase (CdK) inhibitor p21^cip1^ protein with decrease of proteins Cdk2, cyclin E, and Cdk4 level, in a p53-independent mechanism [103]. In nasopharyngeal carcinoma cells, murine squamous cell carcinoma SCCVII, human cancer cell line O12, and canine mammary gland adenocarcinoma cell line the expression of p21^cip1^ and p27^kip1^ were up-regulated whereas the expression of cyclin D1 and phosphorylation of retinoblastoma (Rb) protein was decreased [9]. Interestingly, cell cycle arrest in an oral squamous cell carcinoma could be induced by bLF via down-regulation phospho-p53 and cyclin D1 and up-regulation CdK inhibitor p21 levels [104]. The study by Chea et al. [104] demonstrated that bLF was able to enhance phosphorylation of *p53* in HSC3 cells and induce cell cycle arrest in G0/G1 phase. Activated p53 up-regulate mouse double minute (mdm2) and p21 protein expression, that inhibit cyclin D1 activity [142]. Recently, it has been reported that bLF reflects a novel and unique properties of cell proliferation regulation in Oral squamous cell carcinoma (OSCC) via modulation of mTOR/S6K pathway. bLF downregulates the expression of p-mTOR and in turn p-S6K as well as negatively modulate JAK2/STAT3 signaling pathway via SOCS3 [104].

In addition, it was revealed that bovine LFcin (LFcin-B) blocks cell cycle of CaCo-2 cells, human colon cancer, through prolongation of S phase via down-regulation of cyclin E1 level [105,139]. Whereas delta LF isoform inducts Skp1 (S-phase kinase-associated protein) and Rb (retinoblastoma) genes. Skp1 is a member of Skp1/Cullin-1/F-box ubiquitin ligase complex involved in the proteins ubiquitination and degradation via proteasome at the G1/S transition [147]. 

Furthermore, MDA-MB-231 cells exposed to rhLF (derived from yeast) resulted in the cell cycle arrest at S phase accompanied by percentage reduction of cells at G0/G1 phase and increment of cells at S phase [168]. Similarly, adenocarcinomic human alveolar basal epithelial cells (A549 cells) treated with rhLF but from CHO cells reflects the cell cycle arrest at S phase. Moreover, rhLF exhibits antileukemia selective cytotoxicity, microfilament disruption, cell cycle arrest, and apoptosis activities. rhLF-h-glycan causes cell cycle G2/M arrest and moreover induces DNA fragmentation on ALL cells [167]. Treatment with rhLF (induced in *Aspergillus)* has been shown to block head and neck cancer cell lines at the G0-G1 stage associated with a decrease in the S phase of these cells [107]. The cell cycle arrest is likely mediated by a decrease in phospho-Akt, followed by increase in p27 and a reduction in cyclin E and pRb protein levels, regulating further steps required for DNA synthesis in S phase [107,171].

Moreover recent studies revealed that bLF may inhibit AIEC (adherent-invasive *Escherichia coli*) -mediated genotoxicity through cell cycle manipulation. AIEC strains are able to arrest cell cycle of infected cells in the S phase and induce DNA damage [172].

#### 3.2.6. Apoptosis

Apoptosis, as opposed to accidental necrotic death, is the active death of a cell, and although it can be triggered by the same factors as necrosis, its course is quite different and requires the active participation of the cell in carrying out this suicidal death. This type of death is very beneficial to the organism because it does not lead to development of inflammation and loss of additional cells, but allows for the elimination of cells that are redundant, damaged or threatening to the body, such as cancer cells [149,173]. For it to occur, it requires activation of the expression of genes encoding proteins that lead to the controlled destruction of cellular components, allowing their eventual reuse by surrounding cells or by migrating resident macrophages/phagocytes, which phagocytose the resulting apoptotic bodies containing whole intact cellular organelles derived from the apoptotic cell. Apoptosis is one of the mechanisms that protects against tumorigenesis by triggering the death process in cells undergoing cancer transformation. Unfortunately, many cancer cells have mutations in the genes that trigger/guide the apoptosis process. This leads to the inability of the cell to die and ultimately to the development of a cancerous tumor [174,175]. 

LF has been demonstrated to have both anti-apoptotic [176] or pro-apoptotic effects [114,177]. It may induce cell apoptosis and promote autophagy by regulating the levels of signaling molecules, including: caspase-3 and caspase-8 activation [178,179,180], poly(ADP-ribose) polymerase (PARP) cleavage promotion [179], B-cell lymphoma-2 (Bcl-2) and Bcl-2 associated X (Bax) proteins expression regulation [178,181], increase Fas expression [179], and p53 activation [142], as well as autophagy-related gene 7 (Atg7), Atg12-Atg5 and microtubule-associated protein 1 light chain 3 (LC3)-II/LC3-I [182]. 

Summarize, LF can trigger the Fas signaling or mitochondrial-related or V-H^+^-ATPase-related apoptosis pathway [180,183,184], depending on tissue/cell types [113]. It was shown that bLF induced an increase in the apoptosis of e.g., human breast cancer cell lines HS578T [108], as well as MDA-MB-231 and MCF-7 [185], stomach cancer cells SGC-7901, T cells, as well as highly metastatic prostate adenocarcinoma cell line PC-3 and the osteosarcoma cell line MG-63 containing plasma membrane V-ATPase [186]. The colon tumor cells exposed to bLF reflects increased sensitivity of extrinsic pathway death receptor Fas and activation of caspases-3 and −8 [142]. In stomach cancer cells and squamous cell carcinoma, reduction of protein Bcl-2 level [180] and caspase-3 cleavage [187], respectively, has been observed.

In addition, LF has been shown to activate caspases-3 and -9 but not caspase-8 in leukemic and breast carcinoma cells [112,188]; caspases-3, −7, −8 and −9 in gastric cancer [113], whereas the apoptosis in B-lymphoma has been induced in a caspase-independent fashion [109]. In squamous carcinoma cells activation of caspase-3 and stress activated protein kinases/c-Jun amino-terminal kinase (SAP/JNK) kinase by bLF was revealed [189].

Furthermore, apo-bLF (iron saturated LF) and holo-bLF (low iron saturated LF) were reported to induce apoptosis in both MDA-MB-231 and MCF-7 cells through activation of caspase but via the IAP pathway and the p53 pathway, respectively as well as different affinity [110]. Interestingly, literature data indicate that LF behaves as a V-H^+^-ATPase inhibitor in different model organisms and cell lines [114,117,134,136,190]. bLf selectivity can prevent growth and metastasis of highly metastatic breast and prostate cancer cells, as well as osteosarcoma, which exhibit higher expression levels of plasmalemmal V-H+-ATPase [183,184]. It is worth mentioning that bLF effects on intracellular V-ATPases, was shown too [111,114]. It has also been shown that holo-LF induced ferroptosis, an iron-dependent cell death characterized by cellular accumulation of ROS and lipid peroxidation products, of MDA-MB-231 cells, and promoted ROS generation and DNA damage by radiotherapy and sensitized MDA-MB-231 tumors to radiotherapy [191]. 

On the basis of the assessment of events that depend on V-ATPase function, it has been evaluated that LF can inhibit cell proliferation, induce apoptosis, diminish extracellular acidification rate, and increase intracellular acidification due to inhibition of the proton pumping and ATP hydrolytic activities of V-ATPase [184].

Moreover, lactoferrin-induced apoptotic process in yeast, such as *C. albicans* and *S. cerevisiae*, is one of the element of its antifungal activity [192] as well as bacteria, such as *Lactococcus lactis* and *Pseudomonas aeruginosa* [184]. Literature data indicate that human lactoferrin can induce apoptosis in *Candida albicans* thrush cells, which includes phosphatidylserine transformations, nuclear chromatin condensation, DNA damage and excessive production of reactive oxygen species (ROS) in mitochondria causing their depolarization [137], (Figure 6).

On the other side, LF was found to inhibit the apoptosis in osteoblasts and osteoclasts as well as neutrophils [176]. The anabolic effects of bLF on osteoblasts was observed in vitro in 50–70% of these cells. Recent studies demonstrate that this inhibition involve the insulin-like growth factor (IGF) signaling pathway [176]. Furthermore, the delay of neutrophil apoptosis at sites of infection extend their bactericidal function. The anti-apoptotic effect of lactoferrin is dependent upon its iron saturation status and is mediated at an early stage of apoptosis [193].

#### 3.2.7. Antimicrobial Action

The antibacterial activity of lactoferrin is related, among others, to the ability to bind iron ions, which is required for certain bacteria and fungi growth (*Streptococcus*, *Salmonella*, *Shigella*, *Staphylococcus*, *Enterobacter* and *H. pylori*) as well as biofilm formation [194]. Moreover, lactoferrin by binding to fimbrial adhesins of bacteria prevents adhesion of pathogens to epithelial cells of the host intestines. Blocking the stage of bacterial adhesion on the cell surface prevents them from getting inside the cell and further stages of infection [6]. bLF inhibits the growth of intestinal pathogens, particularly bacteria of the family *Enterobacteriaceae*, as well as stimulates the growth of intestinal microbiota of the genus *Bifidobacterium* or *Lactobacillus* serving as iron donor [194]. It also contributes to the death of bacteria by binding to the bacterial cell wall porin protein or LPS triggering the cell walls or membranes destruction due to increasing their permeability [18]. Moreover, the interactions between LF and LPS increases susceptibility of Gram-negative bacteria to lysozyme [118]. bLF is able to increase the susceptibility of bacteria to some antibiotics and thus reduces the therapeutic dose of the drug, increasing its bactericidal activity [122]. bLF can act as a potent protective molecule against bacterial-induced genotoxicity triggered by AIEC [172].

The antiviral effect of lactoferrin is primarily due to binding to the host cell’s membrane glycosaminoglycans and preventing viruses invasion into host cells. So far, the effect of lactoferrin has been confirmed, including herpes virus types 1 and 2, human cytomegalovirus, human immunodeficiency virus (HIV), human papillomavirus (HPV), rotavirus, enterovirus, adenovirus, influenza virus, parainfluenza virus, hepatitis C virus (HCV), hepatitis B virus (HBV) [123,125,127,128] and SARS-CoV-2 [124,126,195,196,197]. LF may contribute to defence against different species of *Candida* by altering its cell wall integrity causing surface blebs formation, and finally cell death. Additionally, LF has been used in synergy with different antifungal drugs against different yeasts such as *C. dubliniensis, C. albicans, C. glabrata,* and *Cryptococcus*, where their effect was potentiated [137]. In addition, the antimicrobial activity of bLF and elimination of oncogenic pathogens such as, *H.*
*pylori*, responsible for the pathology of gastric ulcers, chronic gastritis, gastric adenocarcinoma and lymphoma, is important for cancer prevention [122]. 

Furthermore, it has been shown that both human and bovine LF exhibit DNase and RNase activity because they can bind and hydrolyze nucleic acids, and this process accelerates in the presence of Mg^2+^ and Ca^2+^ ions [6,73]. This catalytic activity of LF alludes to its protective properties against viruses and bacteria i.e., LF-dependent hydrolysis of the foreign components of pathogens. Worth noting are molecular dynamics simulation studies showing that lactoferramine, lactoferricin, and LFchimera, three antimicrobial peptides derived from camel LF, bind to DNA without DNA sequence preference. Electrostatic interactions between the positively charged side chains of the peptides and the phosphate groups of DNA were confirmed, and that the binding of four copies of LFchimera to DNA is sufficient to initiate a hydrogen bond break between two DNA strands [143]. Therefore, LF-derived peptides can be considered as natural antibiotic formulations, directly targeting and destroying bacterial DNA.

## 4. Conclusions

Lactoferrin is a multipotent protein with various properties, which directly and indirectly affects the immune system of mammals, including humans. Scientists see many biotechnological and medical solutions in this protein, treating it as a factor reducing the risk of infections in prosthetics and implantology, and hope in the possibility of developing new, more effective drug therapies with LF as one of the components of dietary supplements that affect immunity and digestive system and human bacterial flora. Research reports that colostrum and LF may be important nutraceuticals in the prevention or treatment of many ailments. An additional advantage of LF are the reports of DNA protective properties, which guarantee further research on this issue and give the perspective of new therapeutic possibilities in many disease processes with damage to genetic material.

## Figures and Tables

**Figure 1 ijms-23-05248-f001:**
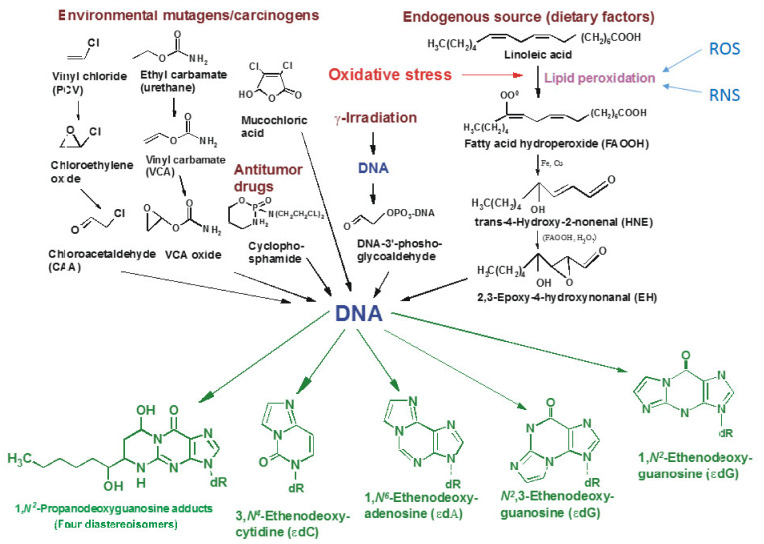
DNA damaging internal and external factors causing formation of etheno and propano base derivatives.

**Figure 2 ijms-23-05248-f002:**
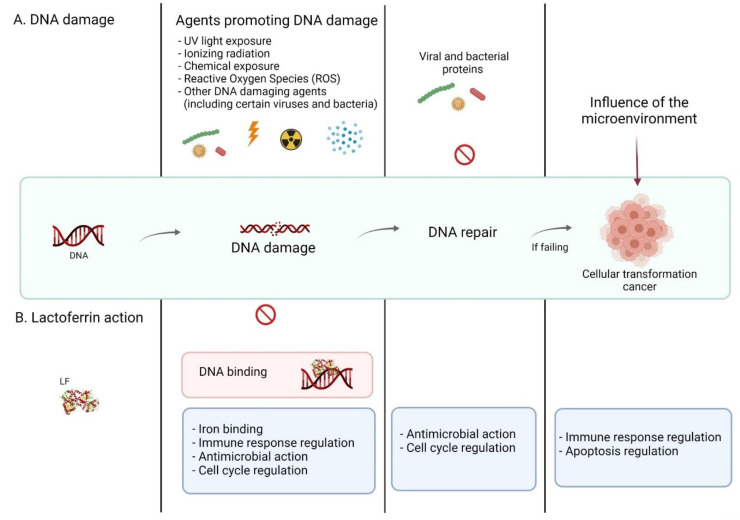
Major factors causing DNA damage and the protective effects of lactoferrin. Factors of external and internal origin as well as bacterial and viral infections can lead to mutations and permanent damage to human genetic material, causing premature aging and cell death, as well as severe impairment of the functions of tissues and organs, which in turn may lead to the development of various diseases, including cancer. In response to these carnivores, cells have developed defense mechanisms to prevent disease and repair damaged genetic material in the form of equal repair systems, such as BER or NER, or by diverting the cell to apoptosis. The protein supporting these mechanisms is Lactoferrin—an endogenous protein which, due to its unique structure, has numerous antibacterial, antiviral, antifungal and anticancer properties. These properties help to bind to damaged (modified) DNA, which supports the functioning of the mechanisms modulating the cell cycle at all its levels and the mechanisms of its repair. Illustration was created in BioRender.com (accessed on 21 April 2022).

**Figure 3 ijms-23-05248-f003:**
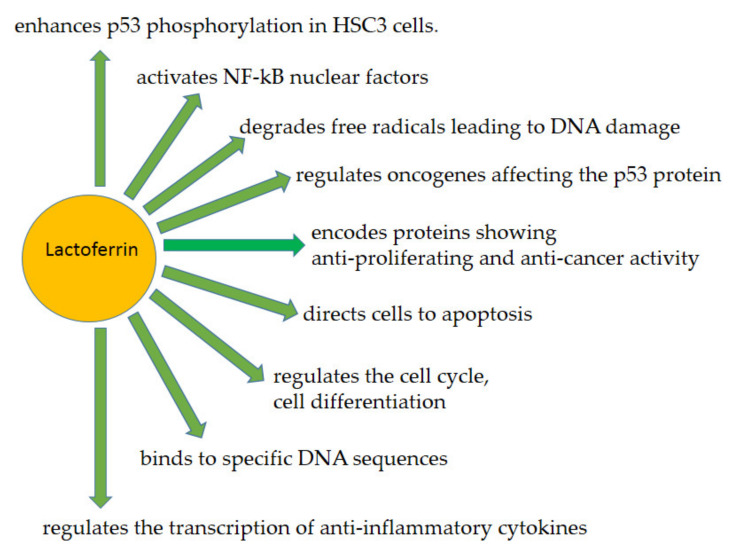
Lactoferrin direct action. LF acts as a multipotent protective factor both by acting on DNA, activating transcription factors, gene expression, regulating the cell cycle, differentiation and leading to the death of cells that threaten the body, such as cancer cells.

**Figure 4 ijms-23-05248-f004:**
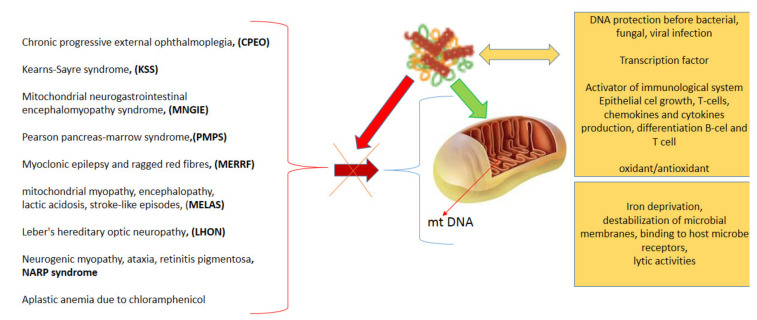
Protective role of LF against mitochondrial damage. Mitochondrial damage is a phenomenon that occurs in various diseases. LF can protect mitochondria by multiple mechanisms, acting as an antipathogenic agent and activating immune cell responses.

**Figure 5 ijms-23-05248-f005:**
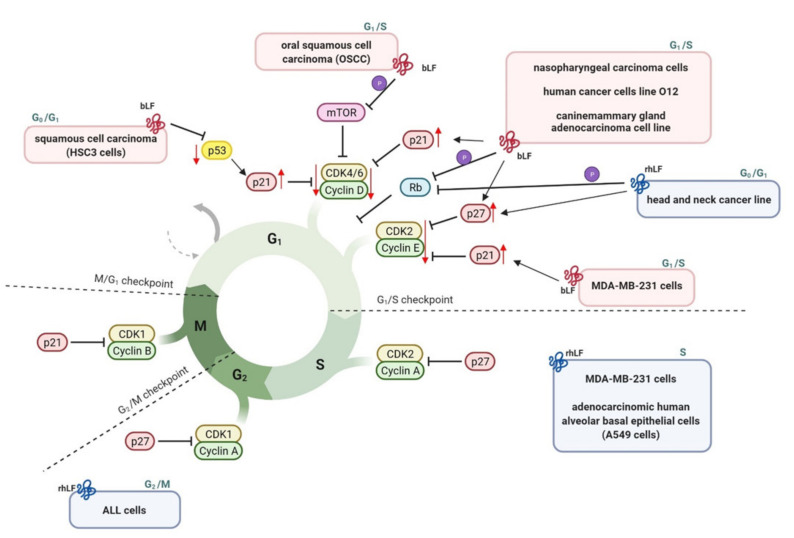
Modulation of cell cycle by bLF and rhLF demonstrated on the example of different cancer cells lines. During the cell cycle, LF can inhibit or activate several checkpoint regulators such as: cyclin-dependent kinase (CDK), and their associated inhibitor partners of the Cip/Kip family proteins (p21, p27), DNA damage response genes (p53) or mTOR/S6K pathway. In response to bLF and rhLF cancer target cells arrest in G_0_/G_1_ phase (e.g., HSC3 cells, head and neck cancer line), G_1_/G_2_ phase (e.g., OSCC, MDA-MD-231 cells, nasopharyngeal carcinoma cells), G2/M phase (e.g., ALL cells) or S phase (e.g., MDA-MB-231 cells, A549 cells). Key molecules that lead to cell cycle arrest are indicated by the red arrows. Illustration was created in BioRender.com (accessed on 21 April 2022).

**Figure 6 ijms-23-05248-f006:**
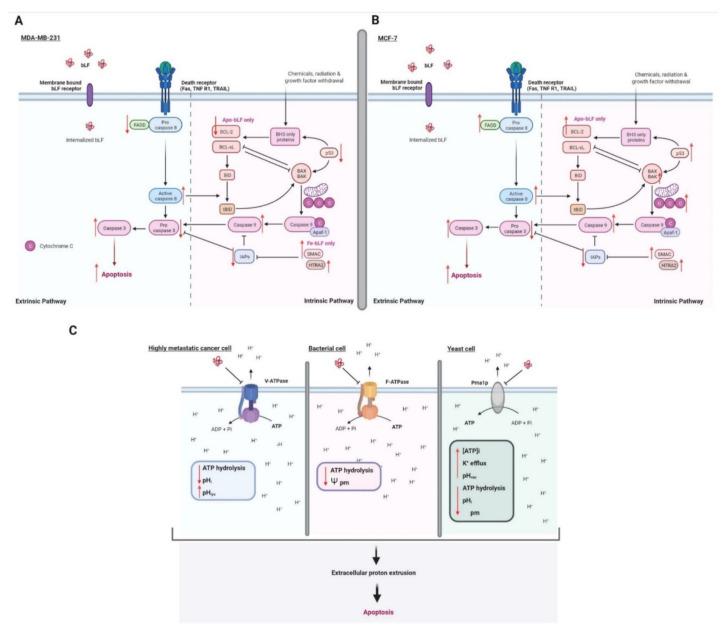
Overview of the mechanism underlying the apoptotic activity of bLF including activation of Fas signaling pathway in MDA-MB-231 cells (**A**), mitochondrial-related pathway in MCF-7 cells (**B**) and inhibition of plasmalemmal V-H+-ATPase proton pumping of highly metastatic cancer cells (V-ATPase), bacteria (F-ATPase) as well as yeast (Pma1p). (**A**,**B**) The internalized, Apo-bLF and Fe-bLF via membrane bound LF receptors in the process of endocytosis, induce or/and inhibit key apoptotic proteins. (**C**) The inhibition of proton efflux trigger extracellular alkalinisation and intracellular acidification. Modulation of key apoptotic molecules level or processes that lead to cell death are indicated by the red arrows. pH_i_: intracellular pH; pH_lys_: lysosomal pH; pH_vac_: vacuolar pH; [ATP]_i_: intracellular ATP concentration; ψ_pm_: plasma membrane potential. Illustration was created in BioRender.com (accessed on 21 April 2022) based on [110,184].

**Table 1 ijms-23-05248-t001:** Lactoferrin properties and mechanism of action—DNA protection in direct and indirect way.

Form of DNA Protection	LF Properties	Mechanism	Ref.
Indirect	Iron saturation	- antioxidant activity (iron chelation) - reduction of oxidative damage (modulation of iron level) - inhibition of bacteria growth	[2,90,91] [6,31,84] [84]
	Immune modulation	- inhibition of pro-inflammatory cytokines production; - stimulation of anti-inflammatory cytokines production; - stimulation of T and B cells maturation; - LPS binding	[24,92,93,94,95] [25,93,96], [4,22,97,98] [33,99,100,101],
	Antitumor	- inhibition of angiogenesis; - iron binding (necessary for cell growth); - inhibition of cell proliferation; - apoptosis induction; - stimulation of lymphocytes, leukocytes and NK activity; - increasing of surface receptors expression on neoplastic cells (identify by the immune system);	[74,102] [79] [103,104,105,106,107] [108,109,110,111,112,113,114] [103,115] [116]
	Antimicrobial	**Antibacterial**: - destruction of bacterial cells (due to: cell wall damage, LPS release or pore proteins binding); - inhibition of bacteria adhesion to the host cells (by degrading bacterial adhesins); - inhibition of biofilm formation; - iron binding (necessary for the growth of bacteria); - increasing the activity of the immune system; - increasing the sensitivity of bacteria to antibiotics; **Antiviral** - inhibition of host cell infection (binding to viral surface proteins or blocking receptors on host cells); - inhibition of viral replication; - stimulation of T lymphocytes to increase the antiviral activity of NK cells; **Antiparasitic** - cell destruction (the cell membrane damage, generation of free radicals); - stimulation of phagocytes activity; - increasing the sensitivity of pathogens to drugs; **Antifungal** - direct cytotoxic effect; - increasing the sensitivity of pathogens to drugs; - stimulation of leukocyte activity;	[9,117,118] [119] [120] [121] [79,93] [122] [123,124,125], [126] [125,127] [79,128] [129,130] [131,132] [133] [134] [135] [136,137]
Direct	DNA binding	**Gene expression modulation** - DNA repair - Cell cycle regulation - Apoptosis regulation - Immune response modulation: ✓ the cytotoxic functions of NK and lymphokine-activated killer cells enhancement; ✓ cytokine synthesis modulation **LF degradation by free radicals**	[26,138,139,140,141,142,143] [91]

## Data Availability

Not applicable.

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
