# Peer review of "Lactoferrin as a Human Genome “Guardian”—An Overall Point of View"

_ijms, 2022, doi:10.3390/ijms23095248_

Round 1

Reviewer 1 Report

I recommend for the publication of this manuscript. 

Author Response

Reviewer 1

I recommend for the publication of this manuscript. 

Dear reviewer,

Thank you very much for your favourable assessment.

Reviewer 2 Report

The authors have improved the manuscript following the suggestions required by the reviewers.

However, few adjustments still need to be performed.

1) Some sentences are too long.
In particular, the first sentence of the abstract needs to be divided immediately two or more.

2) Two phrases indicating the objectives are provided in the abstract, in two different parts.

This is kind weird, please consider to remove the first or mixed both.

3) The caption for some figures are too simple, while others are informative.

Please make all of them informative.

These suggestions are included in the attached file.

Author Response

Reviewer 2 - Luis Claudio

The authors have improved the manuscript following the suggestions required by the reviewers.

However, few adjustments still need to be performed.

1) Some sentences are too long.
In particular, the first sentence of the abstract needs to be divided immediately two or more.

2) Two phrases indicating the objectives are provided in the abstract, in two different parts.

This is kind weird, please consider to remove the first or mixed both.

3) The caption for some figures are too simple, while others are informative.

Please make all of them informative.

These suggestions are included in the attached file.

Dear Reviewer,

Thank you very much for the extremely valuable advice and suggestions of the Reviewer, which contributed to the quality and substantive value of the manuscript.

All suggestions are reflected in this revision.

The revised sentences are indicated in red font:

1) We have divided the first sentence into several sentences

2) We have removed the first phrase indicating the purpose of the work

3) We have improved the descriptions of figures to make them more informative

Reviewer 3 Report

The manuscript entitle “Lactoferrin : a human genome” was revised and the version 2 it is an improved form.

Minor comments:

- A final English spelling and grammar of the text is necessary. Some phrases are too long-i.e the first one in the Abstract!

In conclusion, based on my comments the manuscript can be published after the minor corrections mentioned above

Author Response

The manuscript entitle “Lactoferrin : a human genome” was revised and the version 2 it is an improved form.

Minor comments:

- A final English spelling and grammar of the text is necessary. Some phrases are too long-i.e the first one in the Abstract!

In conclusion, based on my comments the manuscript can be published after the minor corrections mentioned above

Dear Reviewer,

Thank you very much for the extremely valuable advice and suggestions of the Reviewer, which contributed to the quality and substantive value of the manuscript.

The English has been corrected and long sentences have been split into shorter phrases (red font).

This manuscript is a resubmission of an earlier submission. The following is a list of the peer review reports and author responses from that submission.

Round 1

Reviewer 1 Report

Major

  1. The section 2. DNA damage is too general, too long, and too broad, not so informative - which interferes the cohesiveness of lactoferrin and its protective role of genomic DNA that the authors tried to cover in this review. Either focusing on specific DNA damage and mechanisms closely interacts with the lactoferrin function or eliminating the entire section 2 including Fig. 1 and Fig. 2 (at least 2A) from this review will read much better.

  1. Lines 106-106, “The Figure 1 shows the most important sets of mechanisms responsible for the repair process of damaged genetic material.” However, the list of DNA repair in Fig. 1 is mixed up with mechanisms, enzymes, pathways and defective diseases, etc. “act in various pathways”, “Other mechanisms” tell readers almost nothing. This is the secondary reason for suggesting the elimination of the entire section 2.

  1. Lines 195-197, To improve this review more interesting, the authors need to digest the mechanism behind lactoferrin exhibiting multiple enzymatic functions including hydrolase, ATPase, phosphatase, and amylase.

  1. Insert references in Table 1 for each lactoferrin function reported previously.

  1. Lines 208-209, Human serum transferrins.…. They have a similar structure… line 218 an example of transferrin is lactotransferrin (lactoferrin). So, actually how many human transferrins and how similar they are?

  1. Line 228. (Lactoferrin) regulates key antioxidant enzymes but specifically what antioxidant enzymes and how?

  1. Lines 240-242. What is “This protein”? “interaction with various cells and molecules (76) but what cells and molecules and how?

In sum, author’s digestion of previous reports is not sufficient, therefore specific information and mechanisms are largely missing throughout the manuscript.  

Author Response

1.The section 2. DNA damage is too general, too long, and too broad, not so informative - which interferes the cohesiveness of lactoferrin and its protective role of genomic DNA that the authors tried to cover in this review. Either focusing on specific DNA damage and mechanisms closely interacts with the lactoferrin function or eliminating the entire section 2 including Fig. 1 and Fig. 2 (at least 2A) from this review will read much better.

In our opinion, the excessive presentation of the description of DNA damage and its repair mechanisms was intentional because of the presentation of the problem faced by lactoferrin. This is the main core that is the starting point for a further description of the action of lactoferrin. Nevertheless, this chapter has been condensed to the most relevant issues.

However, in our opinion, which is in agreement with the opinion of the other reviewers, a detailed description is needed and Figure 2 should remain in the manuscript to provide the background of the described work. We have added a detailed description to Figure 2.

2.Lines 106-106, “The Figure 1 shows the most important sets of mechanisms responsible for the repair process of damaged genetic material.” However, the list of DNA repair in Fig. 1 is mixed up with mechanisms, enzymes, pathways and defective diseases, etc. “act in various pathways”, “Other mechanisms” tell readers almost nothing. This is the secondary reason for suggesting the elimination of the entire section 2.

Figure  1, now it is 3, was moved to the section 2.2. Oxidative stress and repair mechanisms, and now it contains only repair mechanisms.

3.Lines 195-197, To improve this review more interesting, the authors need to digest the mechanism behind lactoferrin exhibiting multiple enzymatic functions including hydrolase, ATPase, phosphatase, and amylase.

This has been rewritten. Nevertheless there is not much literature on this. Only information that LF binds to nucleotides showing in this way ATPase activity, binds to polysaccharides  showing amylase activity.

4.Insert references in Table 1 for each lactoferrin function reported previously.

We have inserted all the citations on the basis of which it was written right next to the table description.

5.Lines 208-209, Human serum transferrins.…. They have a similar structure… line 218 an example of transferrin is lactotransferrin (lactoferrin). So, actually how many human transferrins and how similar they are?

 There is one plasma transferrin. The text has been corrected. Thanks for catching the error.

  1. Line 228. (Lactoferrin) regulates key antioxidant enzymes but specifically what antioxidant enzymes and how?

 This information has been added.

  1. Lines 240-242. What is “This protein”? “interaction with various cells and molecules (76) but what cells and molecules and how?

This fragment was rearranged to answer your question.

In sum, author’s digestion of previous reports is not sufficient, therefore specific information and mechanisms are largely missing throughout the manuscript.

 We have tried to complete the text and thank you for your valuable suggestion.

Reviewer 2 Report

Good review on "Lactoferrin: a human genome “guardian” I would recommend publication of this review.

Authors should check for grammatical errors in the review before publishing. 

Author Response

Reviewer 2

Good review on "Lactoferrin: a human genome “guardian” I would recommend publication of this review.

Authors should check for grammatical errors in the review before publishing.

Thank you very much for such a positive review. We have made grammatical and language changes

Reviewer 3 Report

The review manuscript from Bukowska-OÅ›ko et al. discussed the role of lactoferrin as an antioxidant agent able to protect the DNA from oxidative damage. The manuscript provides an overview of DNA damage and its consequences. It also covers the several aspects related to lactoferrin structure and properties.

The authors made a good job and their manuscript is well written and organized. It deserves be accepted for publication in this journal after minor adjustments.
I have included my suggestions and corrections in the pdf file attached in this reviewer forum.

Author Response

-Thank you for your insightful review and valuable suggestion. Required changes were made to the text of the manuscript

Reviewer 4 Report

The aim of the authors was to present a review entitled Lactoferrin: a human genome “guardian”. Writing a review it is a big work which involves bringing new and important data related to the subject-matter addressed. Unfortunately it is not the case!

It is hard to consider the manuscript original. The paper does not present any significant progress in the field. It is only a description of DNA damage process and Lactoferrin biological functions. Very little about the role of Lactoferrin as DNA protective factor. Also the References are not updated- i.e the papers by Zheng Zhang et al., Theranostics. 2021; Pan S et al., Int J Oncol. 2021; Soboleva SE et al., J Mol Recognit. 2019; Pirkhezranian Z., et al. BMC Genomics, 2020 are not discussed.

Additionally, the level of scientific depth is low.

In conclusion, although the domain is important, the paper as presently written is unacceptable for publications and need major revisions. The manuscript must be re-submitted and re-reviewed.

Author Response

The aim of the authors was to present a review entitled Lactoferrin: a human genome “guardian”. Writing a review it is a big work which involves bringing new and important data related to the subject-matter addressed. Unfortunately it is not the case!

It is hard to consider the manuscript original. The paper does not present any significant progress in the field. It is only a description of DNA damage process and Lactoferrin biological functions. Very little about the role of Lactoferrin as DNA protective factor. Also the References are not updated- i.e the papers by Zheng Zhang et al., Theranostics. 2021; Pan S et al., Int J Oncol. 2021; Soboleva SE et al., J Mol Recognit. 2019; Pirkhezranian Z., et al. BMC Genomics, 2020 are not discussed.

Additionally, the level of scientific depth is low. In conclusion, although the domain is important, the paper as presently written is unacceptable for publications and need major revisions. The manuscript must be re-submitted and re-reviewed.

  • The intention of the authors of the manuscript was to create a compendium of knowledge on the presented topic. The description of DNA damage and its repair mechanisms was intentional in view of the presentation of the problem faced by lactoferrin. It was a starting point for further description of lactoferrin action. However, the article has been rewritten, additional information has been given. All suggested papers have been discussed and cited.

Round 2

Reviewer 4 Report

The manuscript entitle “Lactoferrin : a human genome” was revised and the version 2 it is an improved form.

Minor comments:

- Figure 2B should be moved to Section 3.

- A final English spelling and grammar of the text is necessary. Some phrases are too long-i.e the first one in the Abstract!

In conclusion, based on my comments the manuscript can be published after the minor corrections mentioned above.

Author Response

The manuscript entitled “Lactoferrin : a human genome” was revised and the version 2 it is an improved form.Minor comments:

1.Figure 2B should be moved to Section 3.

- Figure 2 (now Figure 3) has been moved to section 3. We did not separate this figure because we think the components A and B make a whole.

  1. A final English spelling and grammar of the text is necessary. Some phrases are too long-i.ethe first one in the Abstract! In conclusion, based on my comments the manuscript can be published after the minor corrections mentioned above.

- English corrections according to the reviewer's instructions were made (changes in red color).

Thank for again for reviewing and accepting our paper.

With kind regards

Paweł Kowalczyk
